# Tight Competitive and Variance Analyses of Matching Policies in Gig Platforms

## ABSTRACT

The gig economy features dynamic arriving agents and on-demand services provided. In this context, instant and irrevocable matching decisions are highly desirable due to the low patience of arriving requests. In this paper, we propose an online-matching-based model to tackle the two fundamental issues, matching, and pricing, existing in a wide range of real-world gig platforms, including ride-hailing (matching riders and drivers), crowdsourcing markets (pairing workers and tasks), and online recommendations (offering items to customers). Our model assumes the arriving distributions of dynamic agents (e.g., riders, workers, and buyers) are accessible in advance, and they can change over time, which is commonly referred to as *Known Adversary Distributions* (KAD).

In this paper, we initiate variance analysis for online matching algorithms under KAD. Unlike the popular competitive-ratio (CR) metric, the variance of online algorithms' performance is rarely studied due to inherent technical challenges, though it is well linked to robustness. We focus on two natural parameterized sampling policies, denoted by $\text{ATT}(\gamma)$ and $\text{SAMP}(\gamma)$, which appear as foundational bedrock in online algorithm design. We offer rigorous competitive ratio (CR) and variance analyses for both policies. Specifically, we show that $\text{ATT}(\gamma)$ with $\gamma \in [0, 1/2]$ achieves a CR of $\gamma$ and a variance of $\gamma \cdot (1 - \gamma) \cdot B$ on the total number of matches with $B$ being the total matching capacity. In contrast, $\text{SAMP}(\gamma)$ with $\gamma \in [0, 1]$ accomplishes a CR of $\gamma(1 - \gamma)$ and a variance of $\overline{\gamma}(1 - \overline{\gamma}) \cdot B$ with $\overline{\gamma} = \min(\gamma, 1/2)$. All CR and variance analyses are tight and unconditional of any benchmark. As a byproduct, we prove that $\text{ATT}(\gamma = 1/2)$ achieves an optimal CR of $1/2$.

## CCS CONCEPTS

• **Theory of computation** → **Online algorithms**; • **Applied computing** → *Marketing*.

## KEYWORDS

Online Matching, Competitive Analysis, Variance Analysis

**ACM Reference Format:**
Anonymous Author(s). 2018. Tight Competitive and Variance Analyses of Matching Policies in Gig Platforms. In *Proceedings of The Web Conference (TheWebConf)*. ACM, New York, NY, USA, 14 pages. https://doi.org/XXXXXXX.XXXXXXX

## 1 INTRODUCTION

Markets in the gig economy feature dynamic arriving agents that are typically called *online* as opposed to static agents called *offline* and an instant decision-making requirement due to the low patience of online agents. Examples of gig markets include ride-hailing platforms, crowdsourcing markets, and online recommendations. There are two main categories of studies for gig platforms. The first focuses on the matching issue regarding how to pair up online and offline agents in the system, say, *e.g.*, matching (online) riders and (offline) drivers in ride-hailing [25, 30, 37], pairing (online) workers and (offline) tasks in crowdsourcing markets [13, 14], and offering (offline) items to (online) customers in online recommendations [18]. These works aim to design an efficient matching policy to facilitate as many matches as possible. The second considers another one, called pricing, [4–6, 22, 26, 38], where they typically assume dynamic arriving agents have known or unknown value distributions and we need to design a pricing strategy to promote the total profit flowing into the system.

In this paper, we consider the two issues of matching and pricing simultaneously. Specifically, we assume that the arriving distributions of dynamic agents (*e.g.*, riders in ride-hailing, workers in crowdsourcing, and buyers in online recommendations) and their value functions toward prices are all accessible in advance as part of the input. Our assumptions are inspired by the fact that we can typically learn and estimate these distributions by applying powerful machine learning techniques to massive historical data [9, 19, 34]. We present our model in detail as follows.

**Matching and Pricing in Gig Economies**. We use a bipartite graph $G = (I, J, E)$ to model the network between a set of offline (static) agent types $I$ and a set of online (dynamic) agent types $J$, where an edge $e = (i, j)$ indicates the feasibility of matching agents between types $i$ and $j$ due to practical constraints. Examples include spatial and temporal constraints between a driver (of type $i$) and a rider (of type $j$) in ride-hailing, and the potential interest of a customer $j$ toward an item $i$ in online recommendations. We have a groundset of prices, denoted by $\mathcal{A} = \{a_k | k \in [K] = \{1, 2, \ldots, K\}\}$. We refer to each tuple $f = (i, j, k)$ with $(i, j) \in E$ as an assignment, which represents to match an online agent $j$ with an offline agent $i$ and charge $j$ at the price of $a_k$. Thus, our assignment consists of two parts: matching and pricing. For each offline agent $i \in I$, it has a capacity $b_i \in \mathbb{Z}_+$ representing an upper bound on the number of matches involving $i$. This captures matching caps imposed on $i$, which can be interpreted as the total number of drivers of type $i$ featuring a specific working location in rideshare, the total amount of item $i$ in stock in online recommendations, *etc.*

The online arrival process is as follows. We have a time horizon of $T$ rounds and during each round (or time) $t \in [T] := \{1, 2, \ldots, T\}$, one single online agent $\hat{j}$ will be sampled from $J$ such that $\Pr[\hat{j} = $

$j] = q_{j,t}$ with $\sum_{j \in J} q_{j,t} = 1$ (called $j$ *arrives at $t$*).[1] Let $\mathcal{F} = \{f = (i, j, k) | (i, j) \in E\}$ be the collection of all feasible assignments. For each $f = (i, j, k) \in \mathcal{F}$ and $t \in [T]$, it is associated with a probability $p_{f,t} \in (0, 1]$ and a non-negative profit $w_{f,t}$, which denote the chance of online agent $j$ accepts offline agent $i$ with the price of $a_k$ at time $t$ and the total profit flowing into the system, respectively. Upon the arrival of online agent $j$ at $t$, we (as the algorithm) should either reject $j$ or select an assignment $f = (i, j, k)$ involving $j$. Our decision is instant and irrevocable, and it is then followed by a Bernoulli random choice of $j$ toward the assignment $f = (i, j, k)$: with probability $p_{f,t}$, $j$ accepts $f$ (then $i$'s capacity gets reduced by one and we get a profit of $w_{f,t}$); and with probability $1 - p_{f,t}$, $j$ rejects $f$ and goes away.[2]

An input instance can be characterized as
$\mathcal{I} = \{G = (I, J, E), \mathcal{A}, \{b_i\}, T, \{q_{j,t}\}, \{p_{f,t}, w_{f,t}\}\}$, whose information is all accessible to the algorithm. *Our goal is to design a matching-and-pricing policy such that the total profit is maximized.* Observe that we allow the arrival distributions of online agents to change over time, which is commonly referred to as *Known Adversarial Distributions* (KAD) [12, 40]. In light of this, we refer to our model as *Matching and Pricing under Known Adversarial Distributions* (MP-KAD).

**Remarks on the Sources of Randomness in** MP-KAD. There are three sources of randomness in total: the first is random arrivals of online agents (**Q1**); the second is potentially randomized choices of assignments made by a policy (**Q2**); the third is random acceptance/rejection decisions from online agents over assignments presented by a policy (**Q3**). We assume (**Q1**) and (**Q3**) are independent of each other. Also, the arriving distributions of online agents in (**Q1**) and their Bernoulli random choices in (**Q3**) are all independent over time.

## 1.1 Preliminaries

**Competitive Ratio (CR)**. CR is a metric commonly used to evaluate the performance of online algorithms. Consider a given (online) policy (or algorithm) ALG and a clairvoyant optimal (OPT-OFF). Note that ALG is subject to the real-time decision-making requirement, *i.e.,* ALG has to make an irrevocable decision upon every arrival of online agents before observing future arrivals. In contrast, OPT-OFF is exempt from that requirement: it enjoys the privilege of observing the full arrival sequence of online agents *before* optimizing decisions. Consider a given instance $\mathcal{I}$ of MP, and let E[ALG] and OPT-OFF denote the *expected* total profit achieved by ALG and OPT-OFF, respectively. We say ALG achieves a CR of at least $\rho \in [0, 1]$ if $E[\text{ALG}] \geq \rho \cdot \text{OPT-OFF}$ for all possible instances $\mathcal{I}$. Essentially, CR captures the gap between a policy and a clairvoyant optimal due to the real-time decision-making requirement imposed on the former.

**Variance Analysis in Online Algorithm Design**. For online optimization problems like MP-KAD, we can run any policy *only once* on any given instance. This highlights the importance of variance analysis in online algorithm design, which offers valuable insights into the robustness of related policies. As pointed out by [41], "the CR-metric reflects only the gap between an online policy (ALG) and a clairvoyant optimal in terms of their *expected* performance: it has no guarantee on the variance or robustness of ALG." Additionally, *we want to stress that variance analysis plays a critical role in risk evaluation, particularly for maximization problems as studied here compared with minimization versions.* For minimization problems (say, minimizing some cost), the information of expectation itself can help us upper bound the risk. Let $X \geq 0$ be the total cost incurred by a policy. By applying Markov's inequality, we can upper bound the risk as $\Pr[X \geq N] \leq E[X]/N$ for any target $N > 0$. However, it is a totally different story for maximization as studied here. A lower bound on the expected profit offers no guarantee on the chance of a disastrous event occuring. Specifically, let $X$ denote the random profit gained by a policy. It is possible that $\Pr[X \leq \epsilon] \geq 1 - \delta$ for a given threshold $0 < \epsilon \ll 1$ and any $0 < \delta < 1$ for whatever given value of $E[X]$ when the variance of $X$ is missing. However, after adding the information of any upper bound on the variance, we can immediately estimate the risk by citing concentration inequalities, *e.g.,* $\Pr[X \leq \epsilon] \leq \Pr[|X - \mu| \geq \mu - \epsilon] \leq \text{Var}[X]/(\mu - \epsilon)^2$ for any $\epsilon < \mu = E[X]$.

Inspired by the work of [41], we focus on the variance on *the total number of successful assignments (scheduled and accepted)*, which is due to the randomness as outlined in (**Q1, Q2 and Q3**); see **Remarks on sources of randomness in** MP-KAD.[3] A detailed discussion on the similarities and differences from the work [41] can be seen in Section 1.3.

## 1.2 Benchmark Linear Program (LP)

For ease of presentation, *we assume WLOG that $b_i = 1$ for all $i \in I$ by creating $b_i$ copies of offline agent $i$ where each has a unit capacity.* Thus, the total capacities $B = \sum_{i \in I} b_i = |I|$. For each assignment $f = (i, j, k) \in \mathcal{F}$, let $x_{f,t}$ be the probability that $f$ is selected in a clairvoyant optimal OPT-OFF during round $t$, which includes the probability that $j$ arrives at $t$ (but excludes that $f$ is accepted or rejected by $j$). For each given $j \in J$ and $i \in I$, let $\mathcal{F}_j = \{f = (i, j, k) | (i, j) \in E\}$ and $\mathcal{F}_i = \{f = (i, j, k) | (i, j) \in E\}$ be collections of assignments involving $j$ and $i$, respectively.

$$\max \sum_{t \in [T]} \sum_{f \in \mathcal{F}} x_{f,t} \cdot p_{f,t} \cdot w_{f,t}, \tag{1}$$

$$\sum_{f=(i,j,k) \in \mathcal{F}_j} x_{f,t} \leq q_{j,t}, \forall j \in J, \qquad \forall t \in [T] \tag{2}$$

$$\sum_{t \in [T]} \sum_{f=(i,j,k) \in \mathcal{F}_i} x_{f,t} \cdot p_{f,t} \leq b_i = 1, \forall i \in I \tag{3}$$

$$0 \leq x_{f,t}, \qquad \forall f \in \mathcal{F}. \tag{4}$$

**Lemma 1.** *The optimal value of LP (1) is a valid upper bound on the performance of a clairvoyant optimal of MP-KAD.*

**Proof.** For ease of notation, we use OPT-OFF to denote both a clairvoyant optimal algorithm and its corresponding performance (*i.e.,* expected amount of profit obtained). For each assignment $f = (i, j, k)$, let $X_{f,t} = 1$ indicate that $f$ is selected by OPT-OFF

---

[1]We can always make it equal by creating a dummy node whose arrival simulates the case of no arrival at $t$.

[2]Generally, the profit $w_{f,t}$ collected by the system from $f = (i, j, k)$ at $t$ can be estimated as the price $a_k$ charged to agent $j$ minus the payoff for agent $i$.

[3]As noted by [41], variations in profits can lead to an unbounded variance in the total profit even for simple deterministic policies; thus, we study an unweighted version to make our problem technically tractable.

**Table 1: Notations used throughout this paper.**

| | |
|---|---|
| $[n]$ | Set of integers $\{1, 2, \ldots, n\}$ for a generic integer $n$. |
| $G$ | Input network graph $G = (I, J, E)$. |
| $I(J)$ | Set of offline static agent types (online dynamic types). |
| $\mathcal{A} = (a_k)$ | Groundset of prices. |
| $f = (i, j, k)$ | An assignment of matching agents (of type) $j$ and $i$ and charging $j$ a price of $a_k$. |
| $T$ | Time horizon. |
| $\mathbf{q}_t = (q_{j,t})$ | Arriving distribution at $t$ with $\sum_{j \in J} q_{j,t} = 1$ for all $t \in [T]$. |
| $p_{f=(i,j,k),t}$ | Probability that agent $j$ accepts $i$ with price of $a_k$ at time $t$. |
| $w_{f,t}$ | Profit associated with assignment $f$ if it is successfully made at $t$. |
| $b_i$ | Matching capacity of agent $i \in I$. |
| OPT-LP | Optimal value of Benchmark LP (1). |
| OPT-OFF | A clairvoyant optimal and its corresponding performance. |
| OPT-ON | An online optimal and its performance (subject to the real-time decision restrictions). |

at $t$ (not necessarily accepted by $j$) with $\mathsf{E}[X_{f,t}] = x_{f,t}$. Thus, the performance can be expressed as OPT-OFF $= \sum_{f \in \mathcal{F}} \sum_{t \in [T]} w_{f,t} \cdot p_{f,t} \cdot \mathsf{E}[X_{f,t}] = \sum_{f \in \mathcal{F}} \sum_{t \in [T]} w_{f,t} \cdot p_{f,t} \cdot x_{f,t}$, which matches the objective of LP (1). To prove Lemma 1, it suffices to show that $\{x_{f,t}\}$ is feasible to LP (1).

Let $Y_{j,t} = 1$ indicate that $j$ arrives at $t$ with $\mathsf{E}[Y_{j,t}] = q_{j,t}$. Observe that for any given $j$ and $t$, $\sum_{f \in \mathcal{F}_j} X_{f,t} \leq Y_{j,t}$. Taking expectation on both sides, we get Constraint (2). For each $f = (i, j, k)$ and $t$, let $Z_{f,t} \sim \mathrm{Ber}(p_{f,t})$ simulate the random choices of acceptance and rejection from $j$ over $f$ at $t$. Note that $\sum_{f=(i,j,k) \in \mathcal{F}_i} \sum_{t \in [T]} X_{f,t} \cdot Z_{f,t} \leq b_i$ due to the matching capacity of $b_i$ on offline agent $i$. Taking expectation on both sides yields Constraint (3). The last constraint is trivial. Thus, we justify the feasibility of $\{x_{f,t}\}$ to LP (1) and establish Lemma 1. □

## 1.3 Main Contributions and Related Work

In this paper, we introduce a stochastic optimization model designed to address two fundamental issues, matching and pricing, which are prevalent in a wide range of real-world matching markets. Our focus is on two *natural* LP-based sampling policies that serve as foundational elements in online algorithm design: One includes attenuations, while the other does not. It is important to emphasize that these two LP-based sampling policies and their variations appear in various contexts within online matching markets, as demonstrated by examples in [11, 12, 16, 27, 41]. *Therefore, our primary technical contributions do not lie in algorithm design but rather in providing rigorous and comprehensive competitive and variance analyses for these two representative policies, which we expect can be generalized to other similar settings.*

THEOREM 1. *[Section 2] There is an LP-based sampling policy* **with** *attenuations parameterized with $\gamma \in [0, 1/2]$, denoted by* ATT$(\gamma)$, *which achieves (1) a competitive ratio (CR) of at least $\gamma$ for MP-KAD; (2) a variance of at most $\gamma \cdot (1 - \gamma) \cdot B$ on the total number of successful assignments, where $B = \sum_{i \in I} b_i$, and (3)* ATT$(\gamma = 1/2)$ *achieves an optimal CR of $1/2$ for MP-KAD. Both competitive and variance analyses are tight for any $\gamma \in [0, 1/2]$, which are irrespective of the benchmark LP.*

THEOREM 2. *[Section 3] There is an LP-based sampling policy* **without** *attenuations parameterized with $\gamma \in [0, 1]$, denoted by*

SAMP$(\gamma)$, *which achieves (1) a competitive ratio (CR) of at least $\gamma \cdot (1 - \gamma)$ for MP-KAD; (2) a variance of at most $\overline{\gamma} \cdot (1 - \overline{\gamma}) \cdot B$ on the total number of successful assignments, where $\overline{\gamma} = \min(1/2, \gamma)$ and $B = \sum_{i \in I} b_i$. Both competitive and variance analyses are tight for any $\gamma \in [0, 1]$, which are irrespective of the benchmark LP.*

Finally, we implement both ATT and SAMP and compare them to several heuristics on a real dataset provided by DiDi, Inc., collected in Haikou, China. Detailed results can be seen in the Appendix.

**Remarks on Theorems 1 and 2.** (1) The tightness of the competitive analysis of ATT$(\gamma)$ unconditional of the benchmark LP means that we can identify an instance of MP-KAD on which ATT$(\gamma)$ achieves a CR *equal* to $\gamma$ for any $\gamma \in [0, 1/2]$, where the ratio is computed directly against the clairvoyant optimal by definition instead of the LP value.[4] Similarly, the tightness of the variance analysis suggests that we can identify an instance of MP-KAD on which ATT$(\gamma)$ achieves a variance *equal* to $\gamma$ on the total number of successful matches for any $\gamma \in [0, 1/2]$. The same interpretation applies to SAMP$(\gamma)$. (2) Our analysis indicates that there exists an instance, where ATT$(\gamma)$ achieves the worst (smallest) CR of $\gamma$ and the worst (largest) variance of $\gamma(1 - \gamma)B$ simultaneously for any $\gamma \in [0, 1/2]$; see the example shown in Figure 2. However, that is not necessarily true for SAMP. The CR worst-scenario instance for SAMP shown in Figure 3 has a very different structure from the variance one as illustrated in Figure 4. (3) The optimality of ATT$(\gamma = 1/2)$ is unconditional of the benchmark LP. In other words, no policy can achieve a CR strictly better than $1/2$ for MP-KAD even compared against the clairvoyant optimal directly; see the instance shown in Figure 1. (4) The value of $\gamma \cdot (1-\gamma)$ strictly increases when $\gamma \in [0, 1/2]$. Thus, for ATT$(\gamma)$, the worst-scenario (WS) competitive ratio and variance both increase when $\gamma \in [0, 1/2]$, which suggests a larger profit could come along with a higher variance and vice versa. For SAMP$(\gamma)$, the same trend applies when $\gamma \in [0, 1/2]$, though the WS variance remains unchanged when $\gamma \in [1/2, 1]$. (5) As demonstrated in Theorems 1 and 2, ATT$(\gamma)$ attains a strictly superior competitive ratio (CR) while maintaining the same variance

---

[4]Note that the latter option yields only a lower bound on CR since the LP value is an upper bound on the clairvoyant optimal by Lemma 1.

**Table 2: Comparison of the competitive ratio (CR) and variance achieved by the two LP-based sampling policies,** ATT **, and** SAMP**, in the two models proposed by [41] (results marked in blue) and in this paper (marked in red). For the validity of the assessment, the results in [41] listed below are obtained when $\Delta = 1$ (the number of resources incurred) since each assignment here could cost one unit of the corresponding offline agent's matching capacity only. The terms "conditionally" and "unconditionally" are with respect to the benchmark LP. The unconditionally optimal CR of $1/2$ for** MP-KAD **contrasts with the conditionally tight CR of $1 - 1/e \sim 0.632$ for the model in [41], primarily due to the strict generalization in the arrival setting from KIID [41] to KAD, as studied here.**

| Algorithms | Range of $\gamma$ | CR Bounds | Variance Bounds | Best CR |
|---|---|---|---|---|
| ATT($\gamma$) | $\gamma \in [0,1]$ | $1 - e^{-\gamma}$ | $(1 - e^{-2\gamma} - 2\gamma e^{-\gamma})T^2$ | $1 - e^{-1}$ 
 Condionally Tight |
|  | $\gamma \in [0, 1/2]$ | $\gamma$ | $\gamma(1 - \gamma) \cdot B$ | $1/2$ 
 Unconditionally Optimal |
| SAMP($\gamma$) | $\gamma \in [0,1]$ | $1 - e^{-\gamma}$ | $(1 - e^{-2\gamma} - 2\gamma e^{-\gamma})T^2$ | $1 - e^{-1}$ 
 Condionally Tight |
|  | $\gamma \in [0,1]$ | $\gamma(1 - \gamma)$ | $\overline{\gamma}(1 - \overline{\gamma}) \cdot B, \overline{\gamma} = \min(1/2, \gamma)$ | $1/4$ 
 Unconditionally Tight |

as SAMP($\gamma$) for any $\gamma \in [0, 1/2]$. However, in practice, SAMP enjoys greater efficiency compared to ATT because it does not require the computation of attenuation factors, which is necessary for ATT.

**Comparison Against the Work of [41]**. We outline the distinctions between our work and that of [41] across three different dimensions. **Models**. Both studies focus on allocation policy design within an online-matching-based framework. However, while they concentrate solely on the matching aspect, our approach encompasses both matching and pricing. Additionally, both models accommodate stochastic arrivals of online agents under *known distributions*. In contrast to the assumption of *Known Identical Independent Distributions* (KIID) in [41], we operate within a more general setting known as *Known Adversary Distributions* (KAD), allowing for dynamic changes in arriving distributions over time.[5] Moreover, the two models introduce randomness differently into the matching process. In [41], randomness arises from the cost realization: Each match $e = (i, j)$ incurs a *random* set of static resources. Any policy can match $e$ only when a sufficient budget remains for every possibly needed resource. Importantly, once matched, $e$ is guaranteed to be accepted by $j$ (as no pricing is involved). In contrast, our model introduces randomness in the two possible outcomes—acceptance and rejection—of an arriving agent toward the price included in an assignment. **Algorithms**. Both papers introduce two parameterized LP-based sampling policies, denoted as ATT and SAMP. The key distinction lies in the presence of attenuations in ATT whereas SAMP does not incorporate them. However, it's worth noting that the two papers propose different benchmark LPs due to variations in their respective models. Additionally, for the policy ATT, debilitation factors are computed precisely from the LP solution, whereas in [41], they are derived through Monte Carlo

simulations.[6] **Results**. Both papers claim to conduct tight competitive ratio (CR) analyses for the two LP-based sampling policies. Notably, in [41], the CR tightness is relative to the corresponding benchmark LP, whereas in our analysis, it is unconditional. Furthermore, our variance bounds eliminate the dependence on the length of the time horizon $T$, which is typically assumed to be $T \gg 1$ and can be far larger than the budget $B$. More differences on the results can be seen in Table 2.

**Other Related Work**. There are quite a few previous works that have studied the matching [24, 30, 36] and pricing issues [3, 8, 17, 35] in matching markets. Unlike our setting here, most of them assume at least part of the input is unknown, say, *e.g.,* arriving distributions of online agents and/or acceptance and rejection probabilities are unknown. As a result, they proposed some machine-learning-based frameworks to manage the learning tasks during matching and pricing, among which reinforcement learning and multi-armed bandits are two of the most common paradigms. Another line of research studies has considered matching and pricing jointly in ride-hailing platforms but under an essentially static setting, where requests are assumed known in advance instead of arriving dynamically [23, 32]. In that case, authors typically utilize integer linear programming to resolve the matching issue.

Matching and pricing have also received significant attention in the Operations Research community. We list a few examples as follows. Özkan and Ward [31] have considered matching policy design for ride-hailing services, where they assume both drivers and riders join and depart the system stochastically, and thus, random sojourn time is allowed for every agent. Mahavir Varma et al. [28] have studied a similar setting featured by a two-sided arrival model, and they mainly utilize MDP-based techniques to address the matching and pricing issues. Vera et al. [39] have investigated online resource allocation but focus on evaluating the performance

---

[5]In practice, KAD (Known Adversary Distributions) is a more realistic arrival setting than KIID (Known Identical Independent Distributions) because the arriving distributions of online agents do exhibit variations over time. Please see Figure 8 in the Appendix for a justification on a real ride-hailing dataset.

[6]Monte-Carlo simulations are widely used to approximate attenuation factors [1, 7, 12, 20, 27]. Our policy ATT features that all attenuation factors can be pre-computed explicitly and exactly, which suggests superiority in efficiency in practice; see details of ATT in Algorithm 1.

in terms of regrets, the difference in the profit between a policy and a Prophet, which is very different from the CR metric here.

## 2 AN LP-BASED SAMPLING POLICY WITH ATTENUATIONS (ATT)

Slightly abusing the notation, we use $\{x_{f,t}\}$ to denote an optimal solution to LP (1). The overall picture of ATT$(\gamma)$ with $\gamma \in (0, 1/2]$ works as follows: During each time $t$, we sample an assignment $f = (i, j, k) \in \mathcal{F}_j$ with probability $(x_{f,t}/q_{j,t}) \cdot (\gamma/\beta_{i,t})$ upon the arrival of online agent $j$, where $\gamma$ represents the target competitive ratio we aim to achieve, and $\beta_{i,t}$ is a pre-calculated attenuation factor. The formal statement of ATT$(\gamma)$ is as follows.

---

**Algorithm 1:** A sampling policy with attenuations for MP-KAD: ATT$(\gamma)$, $\gamma \in [0, 1/2]$.

---

1 **Offline Phase**:
2 Solve LP (1) and let $\{x_{f,t}\}$ be an optimal solution.
3 Compute $\beta_{i,t} = 1 - \gamma \sum_{t' < t} \sum_{f \in \mathcal{F}_i} x_{f,t'} \cdot p_{f,t'}$ for all $i \in I$ and $t \in [T]$.
4 **Online Phase**:
5 **for** $t = 1, \ldots, T$ **do**
6     Let an online agent (of type) $j$ arrive at time $t$.
7     Sample an assignment $f = (i, j, k) \in \mathcal{F}_j$ with probability $\frac{x_{f,t}}{q_{j,t}} \frac{\gamma}{\beta_{i,t}}$.
    `/* At most one assignment will be sampled in`
      `Step (7) since`
      $\sum_{f \in \mathcal{F}_j}(x_{f,t}/q_{j,t}) \cdot (\gamma/\beta_{i,t}) \le \sum_{f \in \mathcal{F}_j} x_{f,t}/q_{j,t} \le 1$,
      `which follows from` $\beta_{i,t} \ge \gamma$ `shown in`
      `Ineq. (5) and Const. (2) of LP (1).`     `*/`
8     **if** $i$ is safe at $t$ *(the capacity of $i$ remains)*, **then**
9         Select the assignment $f$ (*i.e.*, assigning $j$ to $i$ and charging $j$ a price of $a_k$);
10     **else**
11         Reject $f$.

---

### 2.1 Competitive Analysis of ATT$(\gamma)$

THEOREM 3. ATT$(\gamma)$ *with* $\gamma \in [0, 1/2]$ *achieves a competitive ratio of* $\gamma$ *for* MP-KAD.

PROOF. We first justify the validity of ATT by showing that the total sum of probabilities on Step (7) is no larger than one. Note that for any $i$ and $t$,

$$\beta_{i,t} = 1 - \gamma \cdot \sum_{t' < t} \sum_{f \in \mathcal{F}_i} x_{f,t'} \cdot p_{f,t'} \ge 1 - \gamma \ge \gamma, \qquad (5)$$

where the first inequality is due to Constraint (3) with $b_i = 1$, and the second follows from $\gamma \le 1/2$. Thus, $\sum_{f=(i,j,k) \in \mathcal{F}_j} \frac{x_{f,t}}{q_{j,t}} \frac{\gamma}{\beta_{i,t}} \le \sum_{f=(i,j,k) \in \mathcal{F}_j} \frac{x_{f,t}}{q_{j,t}} \le 1$, where the second inequality is due to Constraint (2) in LP (1).

For each $i$ and $t$, let $\mathsf{SF}_{i,t} = 1$ indicate that offline agent (of type) $i$ is *safe* at $t$, i.e., it has one unit capacity at (the beginning of) $t$ before any online actions and $\mathsf{SF}_{i,t} = 0$ otherwise. Let $\alpha_{i,t} = \mathsf{E}[\mathsf{SF}_{i,t}]$ be

the probability that $i$ is safe at $t$. For each assignment $f$, let $\chi_{f,t} = 1$ indicate that $f$ is successfully made at $t$ (scheduled and accepted). We now show by induction on $t \in [T]$ that (**P1**) $\alpha_{i,t} = \beta_{i,t}$ and (**P2**) $\mathsf{E}[\chi_{f,t}] = \gamma \cdot x_{f,t} \cdot p_{f,t}$ for all $i \in I, t \in [T]$, and $f \in \mathcal{F}$. Consider the base case $t = 1$. We see that $\alpha_{i,t} = \beta_{i,t} = 1$ for all $i \in I$. For each $f = (i, j, k)$, let $X_{j,t} = 1$ indicate that $j$ arrives at $t$, $Y_{f,t} = 1$ indicate that $f$ gets sampled at $t$, and $Z_{f,t} = 1$ indicate that $j$ accepts $f$ at $t$. Thus, for $t = 1$,

$$\mathsf{E}[\chi_{f,t}] = \mathsf{E}[X_{j,t} \cdot Y_{f,t} \cdot \mathsf{SF}_{i,t} \cdot Z_{f,t}]$$
$$= q_{j,t} \cdot (x_{f,t}/q_{j,t}) \cdot (\gamma/\beta_{i,t}) \cdot \alpha_{i,t} \cdot p_{f,t} = \gamma \cdot x_{f,t} \cdot p_{f,t}. \qquad (6)$$

Now consider a given $t > 1$ and assume (**P1**) and (**P2**) are valid for all $t' < t$. Consider a given $i \in I$. Note that

$$\alpha_{i,t} = \mathsf{E}[\mathsf{SF}_{i,t}] = 1 - \mathsf{E}\Big[ \sum_{t' < t} \sum_{f \in \mathcal{F}_i} \chi_{f,t'} \Big]$$
$$= 1 - \sum_{t' < t} \sum_{f \in \mathcal{F}_i} \gamma \cdot x_{f,t'} \cdot p_{f,t'} = \beta_{i,t},$$

where the third equality is due to the inductive assumption (**P2**) and the last one follows from the definition of $\beta_{i,t}$. We can verify that Equation (6) remains valid for the given $t$ and all $f$ as long as $\alpha_{i,t} = \beta_{i,t}$ for $t$ and all $i \in I$. Thus, we complete the inductive step on (**P1**) and (**P2**). By linearity of expectation, the total expected profit of ATT$(\gamma)$ is equal to

$$\mathsf{E}[\mathsf{ATT}(\gamma)] = \sum_{f \in \mathcal{F}} \sum_{t \in [T]} w_{f,t} \cdot \mathsf{E}[\chi_{f,t}] = \sum_{f \in \mathcal{F}} \sum_{t \in [T]} w_{f,t} \cdot \gamma \cdot x_{f,t} \cdot p_{f,t},$$

which is a fraction of $\gamma$ of the optimal value of LP (1). By Lemma 1, we claim that ATT achieves a competitive ratio at least $\gamma$. □

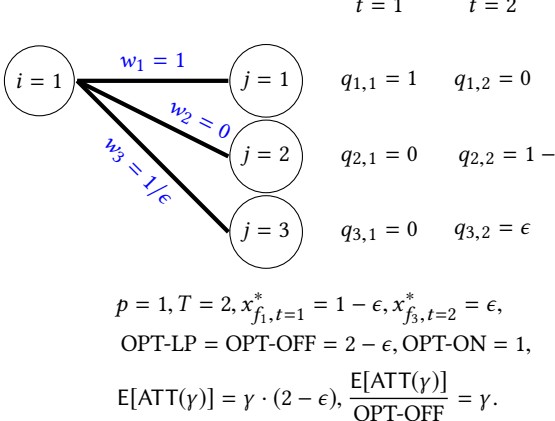

$$p = 1, T = 2, x^*_{f_1, t=1} = 1 - \epsilon, x^*_{f_3, t=2} = \epsilon,$$
$$\text{OPT-LP} = \text{OPT-OFF} = 2 - \epsilon, \text{OPT-ON} = 1,$$
$$\mathsf{E}[\mathsf{ATT}(\gamma)] = \gamma \cdot (2 - \epsilon), \frac{\mathsf{E}[\mathsf{ATT}(\gamma)]}{\text{OPT-OFF}} = \gamma.$$

**Figure 1: An example highlighting the tightness of competitive analysis of ATT$(\gamma)$ for any $\gamma \in [0, 1/2]$ and the optimality of ATT$(\gamma = 1/2)$, where both claims are unconditional of the benchmark LP.**

## 2.2 Tightness of the CR Analysis of $\text{ATT}(\gamma)$ with $\gamma \in [0, 1/2]$ and the Optimality at $\gamma = 1/2$

LEMMA 2. *There exists an instance of MP-KAD such that $\text{ATT}(\gamma)$ achieves a competitive ratio of $\gamma$ for any $\gamma \in [0, 1/2]$. Meanwhile, $\text{ATT}(\gamma = 1/2)$ achieves an optimal competitive ratio of $1/2$ for MP-KAD. Both claims here are irrespective of the benchmark LP.*

EXAMPLE 1. *Consider such an instance as shown in Figure 1. We have one single offline agent with a unit budget that is connected to three online agents indexed by $j = 1, 2, 3$, respectively. Set $T = 2$, and the arriving distributions at $t = 1$ and $t = 2$ are $\mathbf{q}_1 = (1, 0, 0)$ and $\mathbf{q}_2 = (0, 1 - \epsilon, \epsilon)$ with $\epsilon > 0$, respectively. There is one single price, and thus, each edge itself represents an assignment. For ease of notation, we use $j$ to index the three edges and the corresponding assignments, i.e., $f_j = (i = 1, j, *)$ for $j = 1, 2, 3$. Set $p_{f,t} = 1$ for all $f$ and $t$, and the profit on assignments state as follows: $w_{f_1,t} = 1$, $w_{f_2,t} = 0$, and $w_{f_3,t} = 1/\epsilon$ for both $t = 1, 2$.* ∎

PROOF. Consider the example shown in Figure 1. Let OPT-ON, OPT-OFF, and OPT-LP denote the (expected) performance of an online optimal policy, that of a clairvoyant optimal, and the optimal value of Benchmark LP (1) on the above example. We can verify the following facts. First, LP (1) has such an optimal solution that $x^*_{f_1,t=1} = 1 - \epsilon$, $x^*_{f_3,t=2} = \epsilon$ with all rest being zeros, and the corresponding optimal value is OPT-LP $= 2 - \epsilon$. The performance of a clairvoyant optimal is OPT-OFF $= \epsilon \cdot (1/\epsilon) + (1 - \epsilon) = 2 - \epsilon$, while that of an online optimal is OPT-ON $= 1$, which is subject to the real-time decision-making requirements. Second, $\text{ATT}(\gamma)$ with $\gamma \in [0, 1/2]$ achieves a performance of $\gamma \cdot (1-\epsilon) + \gamma \cdot \epsilon \cdot (1/\epsilon) = \gamma \cdot (2-\epsilon)$. Thus, we claim that (1) $\text{ATT}(\gamma)$ achieves a CR of $\gamma(2-\epsilon)/(2-\epsilon) = \gamma$ using the benchmark of either LP (1) or the clairvoyant optimal for any $\gamma \in [0, 1/2]$; and (2) $\text{ATT}(\gamma = 1/2)$ achieves a CR of $1/2$ for MP-KAD, which is optimal since no algorithm can beat the CR of OPT-ON/OPT-OFF $= 1/(2 - \epsilon)$, which approaches $1/2$ when $\epsilon \to 0_+$. □

## 2.3 Variance Analysis of $\text{ATT}(\gamma)$

Recall that for each $f \in \mathcal{F}$ and $t \in [T]$, $\chi_{f,t} = 1$ indicates that $f$ is successfully made (scheduled and accepted) in $\text{ATT}(\gamma)$. Let $H_i = \sum_{f \in \mathcal{F}_i} \sum_{t \in [T]} \chi_{f,t}$ and $H = \sum_{i \in I} H_i$ denote the numbers of successful assignments involving $i$ and in total, respectively. Note that $B = \sum_{i \in I} b_i = |I|$.

THEOREM 4. $\text{Var}[H] \leq \gamma \cdot (1 - \gamma) \cdot B$ with $\gamma \in [0, 1/2]$.

PROOF. For each assignment $f = (i, j, k)$, let $X_{j,t} = 1$, $Y_{f,t} = 1$, and $Z_{f,t} = 1$ indicate that $j$ arrives at $t$, $f$ gets sampled at $t$ in $\text{ATT}(\gamma)$, and $j$ accepts $f$ at $t$, respectively. For each $f = (i, j, k)$, let $W_{f,t} = X_{j,t} \cdot Y_{f,t} \cdot Z_{f,t}$. Note for each given time $t$, we have $\sum_{f \in \mathcal{F}} W_{f,t} \leq \sum_{j \in J} X_{j,t} \leq 1$ since at most one single assignment will be sampled in Step (7) of $\text{ATT}(\gamma)$. Thus, by the Zero-One Principle [15], $\{W_{f,t} | f \in \mathcal{F}\}$ are negatively associated for each given $t$. Observe that $\mathcal{W}_t := \{W_{f,t} | f \in \mathcal{F}\}$ are independent from $\mathcal{W}_{t'}$ as long as $t \neq t'$. As a result, we claim $\{W_{f,t} | f \in \mathcal{F}, t \in [T]\}$ are negatively associated.

Let $\mathcal{W}_i = \{W_{f,t} | f \in \mathcal{F}_i, t \in [T]\}$ for each given $i \in I$. Recall that $H_i = \sum_{f \in \mathcal{F}_i} \sum_{t \in [T]} \chi_{f,t} = 1$ indicates that one assignment in $\mathcal{F}_i$ is *successfully* made in $\text{ATT}(\gamma)$. Since each $i$ has a unit capacity, $H_i = 1$

iff there exists at least one $W_{f,t} \in \mathcal{W}_i$ with $W_{f,t} = 1$. Consequently, $H_i = \min\left(1, \sum_{t \in [T]} \sum_{f \in \mathcal{F}_i} W_{f,t}\right)$, which can be viewed as a non-decreasing function over $\mathcal{W}_i$. Therefore, $\{H_i\}$ can be regarded as a set of non-decreasing functions over disjoint subsets of negatively associated random variables of $\{\mathcal{W}_i | i \in I\}$, which suggests that $\{H_i | i \in I\}$ are also negatively associated [21]. By the result in [33], $\text{Var}[H] = \text{Var}[\sum_{i \in I} H_i] \leq \sum_{i \in I} \text{Var}[H_i]$.

Observe that $H_i$ is a Bernoulli random variable with mean $\mathbb{E}[H_i] = \sum_{t \in [T]} \sum_{f \in \mathcal{F}_i} \mathbb{E}[\chi_{f,t}] = \sum_{t \in [T]} \sum_{f \in \mathcal{F}_i} \gamma \cdot x_{f,t} \cdot p_{f,t} \leq \gamma$, where the second equality is due to Equation (6), while the last inequality due to Constraint (3) in LP (1). Thus, $\text{Var}[H_i] \leq \gamma \cdot (1-\gamma)$ since $\gamma \in [0, 1/2]$. Therefore, we claim that

$$\text{Var}[H] = \text{Var}\left[\sum_{i \in I} H_i\right] \leq \sum_{i \in I} \text{Var}[H_i] \leq \gamma \cdot (1 - \gamma) \cdot B. \quad \square$$

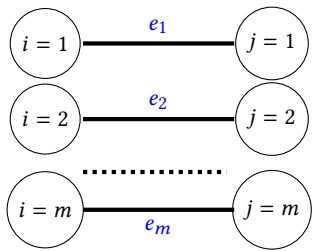

$$|I| = |J| = T = m, p = 1, w = 1, q_{j=t,t} = 1, q_{j\neq t,t} = 0, \forall t \in [T],$$
$$x^*_{i=t,j=t,t} = 1, \forall t \in [T], H = \sum_{i \in I} H_i, H_i = \text{Ber}(\gamma), \forall i \in I,$$
$$\text{Var}[H] = \gamma \cdot (1 - \gamma) \cdot m = \gamma \cdot (1 - \gamma) \cdot B,$$
$$\text{OPT-LP} = \text{OPT-OFF} = m, \mathbb{E}[\text{ATT}(\gamma)]/\text{OPT-OFF} = \gamma.$$

**Figure 2: An example where $\text{ATT}(\gamma)$ achieves the tight CR bound of $\gamma$ and the tight variance bound of $\gamma(1 - \gamma) \cdot B$ simultaneously for any $\gamma \in [0, 1/2]$.**

## 2.4 Tightness of the Variance Analysis of $\text{ATT}(\gamma)$ for any $\gamma \in [0, 1/2]$

LEMMA 3. *There exists an instance of MP-KAD such that $\text{ATT}(\gamma)$ achieves a variance of $\gamma \cdot (1 - \gamma) \cdot B$ on the (random) number of successful assignments for any $\gamma \in [0, 1/2]$.*

PROOF. Consider the instance as shown in Figure 2. We have a graph $G = (I, J, E)$ with $|I| = |J| = |E| = m$, where $E = \{(i, j) | i = j \in [m]\}$ that consists of $m$ edges. There is one single price, and thus, each edge one-one corresponds to an assignment. For ease of notation, we use $e$ to represent the corresponding assignment. Each offline agent type $i$ has a unit capacity $b = 1$. Let $T = m$ and during each time $t \in [T]$, $j = t$ arrives with probability one and no others will arrive. For every assignment $e \in E$ and time $t \in [T]$, $w_{e,t} = p_{e,t} = 1$. We can verify that (1) an optimal solution to LP (1) is as follows: $x^*_{(i,j),t} = 1$ if $i = j = t \in [m]$ and 0 otherwise; (2) the optimal LP value and the performance of a clairvoyant optimal are both $m$. Note that $\text{ATT}(\gamma)$ works as follows: during each round $t \in [T]$ when $j = t$ arrives, $\text{ATT}(\gamma)$ selects the assignment $e = (i = t, j = t)$ with probability $\gamma$ since $\beta_{i=t,t} = 1$.

Let $H_i$ be the number of successful assignments on $i \in I$. We claim that $H_i \sim \text{Ber}(\gamma)$, and thus, $\text{Var}[H_i] = \gamma \cdot (1 - \gamma)$. Observe that $\{H_i\}$ are all independent. Therefore,

$$\text{Var}[H] = \sum_{i \in I} \text{Var}[H_i] = \gamma \cdot (1 - \gamma) \cdot m = \gamma \cdot (1 - \gamma) \cdot B,$$

$$\text{E}[\text{ATT}(\gamma)] = \sum_{i \in I} \text{E}[H_i] = \gamma \cdot m.$$

□

## 3 AN LP-BASED SAMPLING POLICY WITHOUT ATTENUATIONS (SAMP)

In this section, we present another LP-based sampling policy but without attenuations, which is formally stated in Algorithm 2.

---

**Algorithm 2:** An LP-based sampling policy SAMP($\gamma$) for MP-KAD with $\gamma \in [0, 1]$.

**1 Offline Phase**:

**2** Solve LP (1) and let $\{x_{f,t}\}$ be an optimal solution.

**3 Online Phase**:

**4 for** $t = 1, \ldots, T$ **do**

**5**     Let an online agent (of type) $j$ arrive at time $t$.

**6**     Sample an assignment $f = (i, j, k) \in \mathcal{F}_j$ with probability
    $\gamma \cdot x_{f,t}/q_{j,t}$.
    /* At most one assignment will be sampled in
    Step (6) since
    $\sum_{f \in \mathcal{F}_j} \gamma \cdot x_{f,t}/q_{j,t} \le \sum_{f \in \mathcal{F}_j} x_{f,t}/q_{j,t} \le 1$ by
    Const. (2) of LP (1). */

**7**     **if** $i$ is safe at $t$ (i.e., the unit capacity remains), **then**

**8**        Select the assignment $f$ (i.e., assigning $j$ to $i$ and charging $j$ a price of $a_k$);

**9**     **else**

**10**        Reject $f$.

---

### 3.1 Competitive Analysis of SAMP($\gamma$)

THEOREM 5. *SAMP($\gamma$) with $\gamma \in [0, 1]$ achieves a competitive ratio of $\gamma \cdot (1 - \gamma)$ for MP-KAD.*

PROOF. Consider a given $\bar{i}$ and a given time $\bar{t} \in [T]$. We show that $\bar{i}$ is safe at (the beginning of) $\bar{t}$ with a probability of at least $1 - \gamma$. Observe that since $\bar{i}$ has a unit capacity, $\bar{i}$ is safe at $\bar{t}$ iff no successful assignment $f \in \mathcal{F}_{\bar{i}}$ has ever been made before $\bar{t}$.

For each $t < \bar{t}$, let $W_t = \sum_{f=(\bar{i},j,*) \in \mathcal{F}_{\bar{i}}} X_{j,t} \cdot Y_{f,t} \cdot Z_{f,t}$, where $X_{j,t} \sim \text{Ber}(q_{j,t})$, $Y_{f,t} \sim \text{Ber}(\gamma \cdot x_{f,t}/q_{j,t})$, and $Z_{f,t} \sim \text{Ber}(p_{f,t})$ are all Bernoulli random variables simulating the three independent random events, which are $j$ arrives at $t$ ($X_{j,t} = 1$), $f$ gets sampled at $t$ in SAMP ($Y_{f,t} = 1$), and $j$ accepts $f$ at $t$ ($Z_{f,t} = 1$), respectively. Observe that (1) $W_t \in \{0, 1\}$ since at any time $t$, there is at most one single online agent arriving, denoted by $j$, and at most one assignment $f \in \mathcal{F}_j$ gets sampled in Step (6) of SAMP, and (2) $\{W_t | 1 \le t < \bar{t}\}$ are all independent. Let $\text{SF}_{\bar{i},\bar{t}} = 1$ indicate that $\bar{i}$ is

safe at $\bar{t}$. We see that $\bar{i}$ is safe at $\bar{t}$ iff $W_t = 0$ for all $t < \bar{t}$. Thus,

$$\text{E}[\text{SF}_{\bar{i},\bar{t}}] = \Pr\left[\bigwedge_{t < \bar{t}}(W_t = 0)\right] = \prod_{t < \bar{t}} \Pr[W_t = 0]$$

$\left(\text{by independence of } \{W_t | t < \bar{t}\}\right)$

$$= \prod_{t < \bar{t}}\left(1 - \text{E}[W_t = 1]\right) = \prod_{t < \bar{t}}\left(1 - \text{E}\left[\sum_{f=(\bar{i},j,*) \in \mathcal{F}_{\bar{i}}} X_{j,t} \cdot Y_{f,t} \cdot Z_{f,t}\right]\right)$$

$$= \prod_{t < \bar{t}}\left(1 - \sum_{f=(\bar{i},j,*) \in \mathcal{F}_{\bar{i}}} \text{E}[X_{j,t}] \cdot \text{E}[Y_{f,t}] \cdot \text{E}[Z_{f,t}]\right)$$

$\left(\text{by independence of } \{X_{j,t}, Y_{f,t}, Z_{f,t}\}\right)$

$$= \prod_{t < \bar{t}}\left(1 - \sum_{f=(\bar{i},j,*) \in \mathcal{F}_{\bar{i}}} q_{j,t} \cdot (\gamma \cdot x_{f,t}/q_{j,t}) \cdot p_{f,t}\right)$$

$$= \prod_{t < \bar{t}}\left(1 - \sum_{f=(\bar{i},j,*) \in \mathcal{F}_{\bar{i}}} \gamma \cdot x_{f,t} \cdot p_{f,t}\right)$$

$$\ge 1 - \sum_{t < \bar{t}} \sum_{f=(\bar{i},j,*) \in \mathcal{F}_{\bar{i}}} \gamma \cdot x_{f,t} \cdot p_{f,t} \ge 1 - \gamma. \qquad (7)$$

$\left(\text{by Constraint (3) of LP (1)}\right)$

The analysis above suggests that for any given assignment $\bar{f} = (\bar{i}, \bar{j}, *) \in \mathcal{F}_{\bar{i}}$, it is successfully made at $\bar{t}$ by SAMP($\gamma$), denoted by $\chi_{\bar{f},\bar{t}} = 1$, with probability of at least

$$\text{E}[\chi_{\bar{f},\bar{t}}] = \text{E}\left[X_{\bar{j},\bar{t}} \cdot Y_{\bar{f},\bar{t}} \cdot \text{SF}_{\bar{i},\bar{t}} \cdot Z_{\bar{f},\bar{t}}\right] \ge (1 - \gamma) \cdot \gamma \cdot x_{\bar{f},\bar{t}} \cdot p_{\bar{f},\bar{t}}.$$

Therefore, we claim that the expected amount of profit gained by SAMP($\gamma$) should be at least

$$\text{E}[\text{SAMP}(\gamma)] = \sum_{f \in \mathcal{F}} \sum_{t \in [T]} w_{f,t} \cdot \text{E}[\chi_{f,t}] \ge (1 - \gamma) \cdot \gamma \cdot x_{f,t} \cdot p_{f,t} \cdot$$
$$w_{f,t}$$
$$= (1 - \gamma) \cdot \gamma \cdot \text{OPT-LP} \ge (1 - \gamma) \cdot \gamma \cdot \text{OPT-OFF},$$

where OPT-LP refers to the optimal value of LP-(1) and OPT-OFF the performance of a clairvoyant optimal, and the last inequality is due to Lemma 1. We establish the claim that SAMP($\gamma$) achieves a competitive ratio of at least $\gamma \cdot (1 - \gamma)$. □

### 3.2 Tightness of the Competitive Analysis of SAMP($\gamma$) for any $\gamma \in [0, 1]$

LEMMA 4. *There exists an instance of MP-KAD such that SAMP($\gamma$) achieves a competitive ratio of $\gamma \cdot (1 - \gamma)$ for any $\gamma \in [0, 1]$ regardless of the benchmark LP.*

PROOF. Consider the instance as shown in Figure 3, which is almost the same as that in Figure 1 except that the profit on $f_3$ is $w_{f_3,t} = 1/\epsilon^2$ instead of $1/\epsilon$ for both $t = 1, 2$. We can verify the following facts. First, LP (1) has such an optimal solution that $x^*_{f_1,t=1} = 1 - \epsilon, x^*_{f_3,t=2} = \epsilon$ with all rest being zeros, and the corresponding optimal value is OPT-LP $= 1/\epsilon + 1 - \epsilon$. The performance of a clairvoyant optimal is OPT-OFF $= 1/\epsilon + 1 - \epsilon$. Second, SAMP($\gamma$) with $\gamma \in [0, 1]$ samples $f_1$ with probability $\gamma(1 - \epsilon)$ when $j = 1$ arrives at $t = 1$, and it samples $f_3$ with probability $\gamma$ if $j = 3$ arrives at $t = 2$. As a result, the probability that $f_1$ is successfully made is $\gamma(1 - \epsilon)$ and that of $f_3$ is $\epsilon \cdot \gamma \cdot \text{E}[\text{SF}_{i=1,t=2}] = \epsilon \cdot \gamma \cdot (1 - \gamma \cdot (1 - \epsilon)) =$

$\epsilon \cdot \gamma \cdot (1 - \gamma + \gamma \cdot \epsilon)$. This suggests SAMP($\gamma$) gains an expected amount of profit of $\gamma(1 - \epsilon) + (1/\epsilon) \cdot \gamma \cdot (1 - \gamma + \gamma \cdot \epsilon)$. Thus, we claim that SAMP($\gamma$) achieves a CR of $\gamma(1 - \gamma)$ when $\epsilon \to 0_+$ based on the benchmark of either LP (1) or the clairvoyant optimal for any $\gamma \in [0, 1]$. □

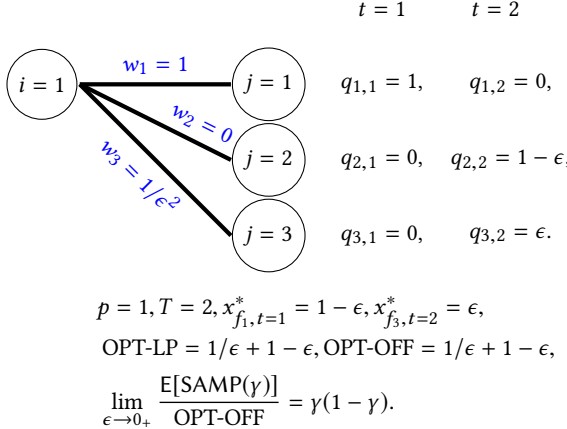

$$p = 1, T = 2, x^*_{f_1, t=1} = 1 - \epsilon, x^*_{f_3, t=2} = \epsilon,$$

$$\text{OPT-LP} = 1/\epsilon + 1 - \epsilon, \text{OPT-OFF} = 1/\epsilon + 1 - \epsilon,$$

$$\lim_{\epsilon \to 0_+} \frac{\text{E}[\text{SAMP}(\gamma)]}{\text{OPT-OFF}} = \gamma(1 - \gamma).$$

**Figure 3: An example highlighting the *unconditional* tightness of competitive analysis of** SAMP($\gamma$) **for any** $\gamma \in [0, 1]$, **which is irrespective of the benchmark LP.**

## 3.3 Variance Analysis of SAMP($\gamma$)

Recall that for each $f \in \mathcal{F}$ and $t \in [T]$, $\chi_{f,t} = 1$ indicates $f$ is successfully made (scheduled and accepted) in SAMP($\gamma$) with $\gamma \in [0, 1]$. Let $H_i = \sum_{f \in \mathcal{F}_i} \sum_{t \in [T]} \chi_{f,t}$ and $H = \sum_{i \in I} H_i$ denote the numbers of successful assignments involving $i$ and in total, respectively.

**Theorem 6.** $\text{Var}[H] \leq \overline{\gamma} \cdot (1 - \overline{\gamma}) \cdot B$, where $\overline{\gamma} = \min(1/2, \gamma)$ and $B = \sum_{i \in I} b_i$ with $\gamma \in [0, 1]$.

**Proof.** For each $f = (i, j, k)$, let $X_{j,t} = 1, Y_{f,t} = 1$, and $Z_{f,t} = 1$ indicate that $j$ arrives at $t$, $f$ gets sampled at $t$ in SAMP($\gamma$), and $j$ accepts $f$ at $t$, respectively. For each $f$ and $t$, let $W_{f,t} = X_{j,t} \cdot Y_{f,t} \cdot Z_{f,t}$. Observe that $\text{E}[W_{f,t}] = \gamma \cdot x_{f,t} \cdot p_{f,t}$. For each $i$ and $t$, let $W_{i,t} = \sum_{f \in \mathcal{F}_i} W_{f,t}$ and $W_i = \sum_{t \in [T]} W_{i,t}$. Following analyses in the proof of Theorem 5, we have (1) $\{W_{i,t} | t \in [T]\}$ are independent for each given $i \in I$, and each $W_{i,t} \in \{0, 1\}$ with $\text{E}[W_{i,t}] = \sum_{f \in \mathcal{F}_i} \gamma \cdot x_{f,t} \cdot p_{f,t}$; and (2) $i$ is not matched in the end, *i.e.*, $H_i = 0$, iff $W_i = 0$. Thus, we see that $H_i = \min(W_i, b_i) = \min(W_i, 1)$. Following similar analyses in the proof of Theorem 4, we can show that $\{W_{f,t} | f \in \mathcal{F}, t \in [T]\}$ are negatively associated, and so are $\{H_i | i \in I\}$. Therefore, $\text{Var}[H] = \text{Var}[\sum_{i \in I} H_i] \leq \sum_{i \in I} \text{Var}[H_i]$.

Consider a given $i \in I$. Observe that $i$ is not matched is equivalent to that $i$ stays safe to the end of $T$, denoted by $\text{SF}_{i,T+1}$. By Inequality (7),

$$\text{E}[H_i] = 1 - \Pr[H_i = 0] = 1 - \text{E}[\text{SF}_{i,T+1}] \leq \gamma,$$

which suggests that $\text{Var}[H_i] \leq \overline{\gamma} \cdot (1 - \overline{\gamma})$ with $\overline{\gamma} = \min(1/2, \gamma)$. Consequently, we have $\text{Var}[H] = \text{Var}[\sum_{i \in I} H_i] \leq \sum_{i \in I} \text{Var}[H_i] \leq \overline{\gamma} \cdot (1 - \overline{\gamma}) \cdot B$. □

## 3.4 Tightness of the Variance Analysis of SAMP($\gamma$) for any $\gamma \in [0, 1]$

**Lemma 5.** *There exists an instance of* MP-KAD *such that* SAMP($\gamma$) *achieves a variance of* $\overline{\gamma} \cdot (1 - \overline{\gamma}) \cdot B$ *on the (random) number of successful assignments for any* $\gamma \in [0, 1]$, *where* $\overline{\gamma} = \min(1/2, \gamma)$.

**Proof.** Consider the instance as shown in Figure 4, which is almost the same as that in Figure 2 except that $p_{f,t} = \min(1, (1/2)/\gamma)$ instead of 1 for all $f$ and $t$. SAMP($\gamma$) works as follows: during each round $t \in [T]$ when $j = t$ arrives, SAMP($\gamma$) selects the assignment $f = (i = t, j = t)$ with probability $\gamma$, which suggests $f$ is successfully made with probability $\gamma \cdot p_f = \overline{\gamma}$.

Let $H_i$ be the number of successful assignments on $i \in I$. We claim that $H_i \sim \text{Ber}(\overline{\gamma})$, and thus, $\text{Var}[H_i] = \overline{\gamma} \cdot (1 - \overline{\gamma})$. Observe that $\{H_i\}$ are all independent, therefore,

$$\text{Var}[H] = \text{Var}\left[\sum_{i \in I} H_i\right] = \sum_{i \in I} \text{Var}[H_i]$$

$$= \overline{\gamma} \cdot (1 - \overline{\gamma}) \cdot m = \overline{\gamma} \cdot (1 - \overline{\gamma}) \cdot B.$$

□

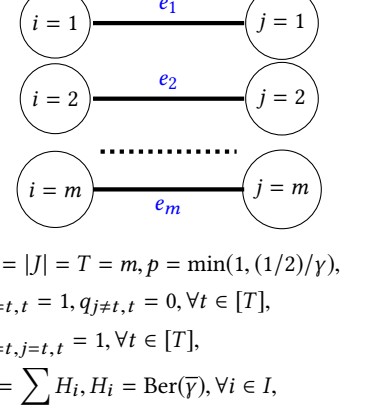

$$|I| = |J| = T = m, p = \min(1, (1/2)/\gamma),$$

$$q_{j=t,t} = 1, q_{j \neq t, t} = 0, \forall t \in [T],$$

$$x^*_{i=t, j=t, t} = 1, \forall t \in [T],$$

$$H = \sum_{i \in I} H_i, H_i = \text{Ber}(\overline{\gamma}), \forall i \in I,$$

$$\text{Var}[H] = \overline{\gamma} \cdot (1 - \overline{\gamma}) \cdot m = \overline{\gamma} \cdot (1 - \overline{\gamma}) \cdot B.$$

**Figure 4: An example highlighting the tightness of variance analysis of** SAMP($\gamma$) **for any** $\gamma \in [0, 1]$.

## 4 CONCLUSION

In this paper, we study matching and pricing simultaneously emerging in various gig platforms. We focus on two fundamental LP-based sampling algorithms and provide tight competitive and variance analyses for each of them. Our research opens a few directions. The immediate one is to extrapolate the current variance-analysis techniques to more general settings such as a weighted objective, *i.e.,* the variance of the total profit instead of the total number of assignments, and less strict arriving assumptions, *e.g.,* random arrival order and adversary. We expect more technical challenges there, which perhaps require us to add extra assumptions to make the problem tractable.

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

# A EXPERIMENTAL RESULTS

**Table 3: Parameter settings, where $|I|$ and $|J|$ denotes the numbers of driver and rider types, $[\underline{b}, \overline{b}]$ denotes the range of capacities among all driver types, $B$ is the total capacities of all driver types (*i.e.,* the total number of drivers over all types), $T$ is the total number of online rounds, $\mathcal{A}$ is set of basic prices per kilometer, and $\gamma$ is the parameter in ATT and SAMP.**

|  | Duration | $|I|$ | $|J|$ | $[\underline{b}, \overline{b}]$ | $B$ | $T$ | $\mathcal{A}$ | $\gamma$ |
|---|---|---|---|---|---|---|---|---|
| General Case | 8:00-20:00 | 50 | 80 | {[1,3],[1,7],[1,11], [1,15],[1,19],[1,23]} | {100,200,...,600} | 4200 | {2.2,2.4, ...,3.2} | {0.1,0.2,...,0.5} |
| Special Case with Large Capacities | | | | {[10,20],[20,30],[30,40], [40,50],[50,60]} | {750,1250,...,2750} | $10 \times B$ | | {0.2,0.4,...,1.0} |

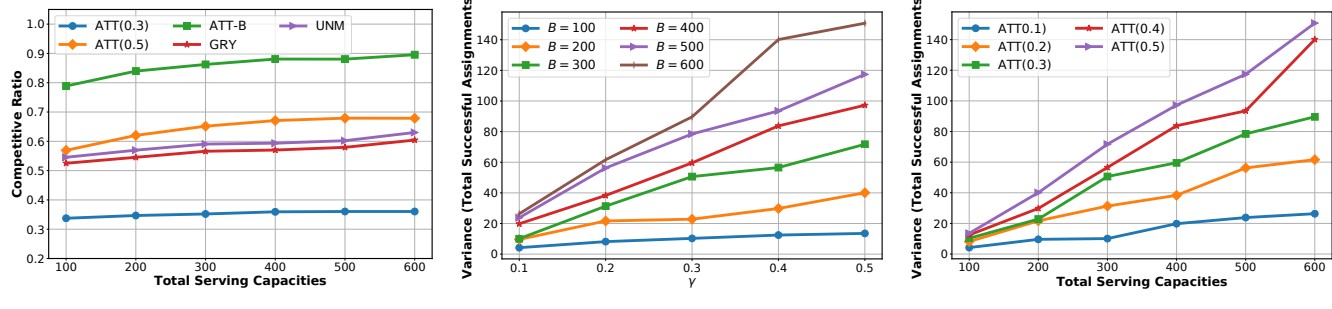

**(a) Competitive ratios when the total serving capacities $B \in \{100, 200, \ldots, 600\}$.**

**(b) Variances of ATT($\gamma$) when $\gamma$ takes values in $\{0.1, 0.2, \ldots, 0.5\}$ for different fixed $B$ values.**

**(c) Variances of ATT($\gamma$) when $B \in \{100, \ldots, 600\}$ with different fixed $\gamma$ values.**

**Figure 5: Results for the general case on a real dataset offered by DiDi, Inc., collected in Haikou, China.**

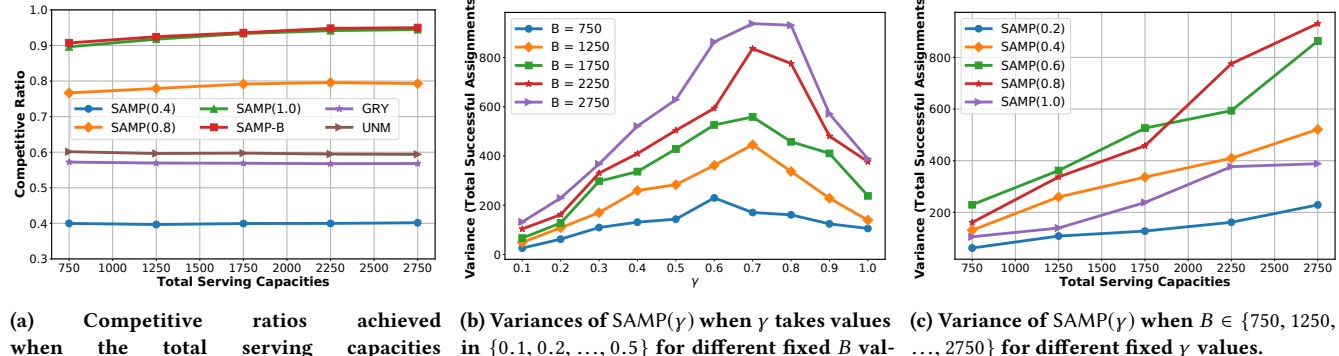

**(a) Competitive ratios achieved when the total serving capacities $B \in \{750, 1250, \ldots, 2750\}$.**

**(b) Variances of SAMP($\gamma$) when $\gamma$ takes values in $\{0.1, 0.2, \ldots, 0.5\}$ for different fixed $B$ values.**

**(c) Variance of SAMP($\gamma$) when $B \in \{750, 1250, \ldots, 2750\}$ for different fixed $\gamma$ values.**

**Figure 6: Results for the special case of large capacities on a real dataset offered by DiDi, Inc., collected in Haikou, China.**

**Data Preprocessing**. We conduct our experiments on a real-world ride-hailing dataset offered by DiDi Inc., which is collected in Haikou, China. Each trip record consists of 23 columns, including pick-up and drop-off locations, the timestamp when the trip started and ended, and the estimated total fare for the order, *etc.* Following the setting of work [29], we set drivers offline agents while riders are online agents that arrive dynamically. We focus on the case where longitude and latitude range in (110.18, 110.48) and (19.90, 20.10), respectively, and we partition the area into $15 \times 10 = 150$ grids with equal size.

We construct the input instance as follows. Focus on the time window from 8:00 to 20:00 on May 5, 2017. We choose $T = 4200$ and split the total 12 hours in the window into $T$ intervals such that each spans 30 seconds. For each grid, we create a driver type $i$, and sample a serving capacity $b_i$ from $[\underline{b}, \overline{b}]$ uniformly at random, which denotes the number of drivers of type $i$ in the system. For each pair of starting

and ending grids, we create a rider type $j$, and set its arriving probability at time $t$ (i.e., $q_{j,t}$) as the proportion of its record number among all rider types during $t$. We select the top 50 driver types and 80 rider types in terms of their number of records. For each pair of driver-rider types, we add an edge with probability 1 if the two share the same starting grid; with probability 1/4 if they are adjacent to each other.

In our context of ride-hailing services, the groundset of prices applying to all ride types can be huge due to the large variance in trip length. Instead, we introduce the groundset of basic prices per kilometer (denoted by $\mathcal{A}$ as well) and for each $a_k \in \mathcal{A}$, the assignment $f = (i, j, k)$ should be re-interpreted as assigning $j$ to $i$ and charge $j$ at a price of $a_k$ per kilometer (and thus, the total price charged to $j$ should be equal to $a_k$ multiplied by its trip length). For each given rider type $j$ and time $t$, let $F_{j,t}$ be the CDF (Cumulative Distribution Function) of private value distribution of $j$ over basic prices at $t$, i.e., $F_{j,t}(a_k)$ denotes the probability that the expected price (per kilometer) from $j$ is no larger than $a_k$ at time $t$. Commonly, the (random) private value is set to follow either a normal distribution [38, 42] or an exponential distribution [6, 26]. In our case, we consider a truncated normal distribution for each $F_{j,t}$, where the mean and variance are estimated based on samples of records relevant to rider type $j$ at $t$. In this way, for each assignment $f = (i, j, k)$, we set the acceptance probability of $j$ accepting $f$ as $p_{f,t} = 1 - F_{j,t}(a_k)$, i.e., the probability that ride type $j$ holds a private value larger or equal to the given price $a_k$. For each $f = (i, j, k)$, we set $w_{f,t} = \lambda \cdot a_k \cdot \kappa_j$, where $\lambda$ refers to the royalty rate and $\kappa_j$ is the trip length of rider type $j$. We set $\lambda$ to 25% by default [2].

**Algorithms**. In addition to the two LP-based algorithms ATT and SAMP proposed in this paper, we implement several baselines as follows. Consider a rider of type $j$ arrives at time $t$, and let $\mathcal{F}_{j,t}$ be the collection of *safe* assignments with respect to $j$ at $t$ (note that $\mathcal{F}_{j,t}$ may get reduced as time). Recall that $\{x_{f,t}\}$ is an optimal solution to the benchmark LP (1). If $\mathcal{F}_{j,t}$ is empty, then we have to reject $j$. Otherwise, (a) ATT-B (a boosted version of ATT): Samples an assignment $f = (i, j, k) \in \mathcal{F}_{j,t}$ with probability $\frac{x_{f,t}}{q_{j,t} \cdot \beta_{i,t}} / \sum_{f \in \mathcal{F}_{j,t}} \frac{x_{f,t}}{q_{j,t} \cdot \beta_{i,t}}$; (b) SAMP-B (a boosted version of SAMP) : Sample an assignment $f = (i, j, k) \in \mathcal{F}_{j,t}$ with probability $\frac{x_{f,t}}{q_{j,t}} / \sum_{f \in \mathcal{F}_{j,t}} \frac{x_{f,t}}{q_{j,t}}$; (c) GRY: Select an assignment $f \in \mathcal{F}_{j,t}$ that maximizes $p_{f,t} \cdot w_{f,t}$ (and break ties arbitrarily); (d) UNM: Sample an assignment in $\mathcal{F}_{j,t}$ uniformly at random.

**Computational Complexity of** ATT **and** SAMP. Both ATT and SAMP consist of two parts: **Offline Phase** and **Online Phase**. As for **Offline Phase**, both ATT and SAMP need to solve the benchmark LP (1) that has $N := |E| \cdot K \cdot T$ variables. Thus, theoretically the running time on the part of solving LP (1) can be as low as $O^*(N^{2+1/6} \log(N/\delta))$ [10], where $\delta$ is the relative accuracy and $N = |E| \cdot K \cdot T$ with $K$ and $T$ being the total number of prices and online rounds, respectively. For SAMP, **Offline Phase** involves one extra procedure of computing all attenuation factors $\{\beta_{i,t}\}$ that takes another $O(T \cdot |E| \cdot K)$. As for **Online Phase**, both ATT and SAMP just need to sample an assignment from a one-dimensional vector with a size no larger than $|I| \cdot K$, which takes $O(|I| \cdot K)$ time. Thus, the dominant part of the running time will be solving the benchmark LP (1) in **Offline Phase**. Fortunately, all computations in **Offline Phase** can be done well before the online process starts.

**Results and Discussions**. For each instance, we run all algorithms for 100 times and take the average as the final performance (the total expected profits obtained). We compute the ratio of the performance of each algorithm to the optimal value of LP (1) as the final competitive ratio achieved. Additionally, we output the total number of successful assignments (that are scheduled and accepted) and the resulting variance.

Figure 5a shows the performance of ATT($\gamma$) is quite stable: its competitive ratios always stay slightly above and almost match the theoretical lower bounds of $\gamma$. This confirms our theoretical prediction in Theorem 1 and highlights the tightness of our competitive-ratio analysis. Though ATT($\gamma$) with $\gamma = 1/2$ proves optimal in the worst-case, its practical performance seems not as good as the boosted version ATT-B. This is mainly due to the fact that real-world instances deviate largely from the worst-case version. Figure 5a suggests that when the total number of arrivals of riders is fixed ($T = 4200$), heuristics like GRY and UNM are dominated by ATT-B for any given $B$, showing that ATT-B can work well in practice. In addition, ATT(0.5) has a prominent advantage over the two heuristics when $B$ is moderate and still slightly outperforms them even when $B$ is extremely small or large. In contrast, Figure 5c suggests that the variance increases almost linearly as $B$ for each fixed $\gamma$. These results align perfectly with our theoretical upper bounds in Theorem 1.

As for the special case with large capacities, Figure 6a shows that as $\gamma$ increases, the competitive ratios of SAMP($\gamma$) will increase as well. This confirms our theoretical prediction in the first part of Theorem 2. Moreover, SAMP(0.8) and SAMP(1.0) always outperform GRY and UNM over different choices of $B$. Although GRY and UNM perform well in the metric of total successful assignments, they fail to achieve a good competitive ratio finally. This is mainly due to the myopia of heuristic-based strategies: they will run out the supplies very quickly and lose the opportunities to serve those profitable demands in the future. On the other hand, the LP-based algorithms will optimize the usage of supplies globally, which results in an extremely high competitive ratio overall. Figure 6c shows that the variance for all instances is upper bounded by $\bar{\gamma} \cdot (1 - \bar{\gamma}) \cdot B$, as suggested in the second part of Theorem 2.

# B GRID PARTITION IN HAIKOU, CHINA.

In our experiments, we focus on the area where longitude and latitude range in (110.18, 110.48) and (19.90, 20.10), respectively, and we partition the area into $15 \times 10 = 150$ grids with equal size, as shown in Figure 7.

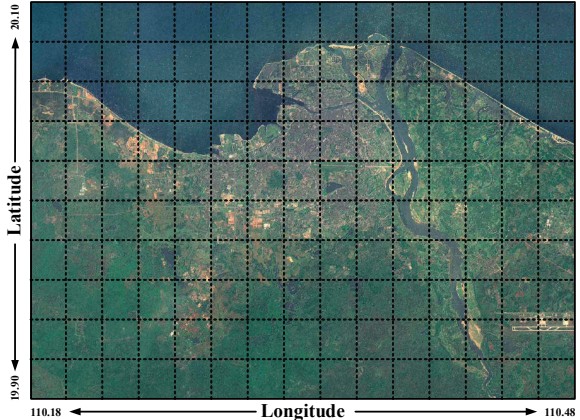

**Figure 7: Grid partition in Haikou, China.**

## C    JUSTIFICATION OF ASSUMPTIONS IN THE MODEL

First, we validate our KAD (Known Adversary Distributions) arrival assumption by plotting the distribution of the total number of arrivals for all rider types at various times; see Figure 8. The results demonstrate that the arrival patterns of each rider type do indeed change significantly over time. Notably, there is a peak-hour trend between 16:00 and 18:00 when most rider types experience their highest arrival rates.

Second, we validate the dependency of the acceptance probability on rider types and time by plotting the distribution of the number of trip records at different times for each given rider type and by plotting the distribution of the number of trip records for different rider types within each predefined time interval, as illustrated in Figure 9 and Figure 10. In Figure 10, it's evident that for a given time slot between 16:00 and 18:00, acceptance prices are partially concentrated around 3.0 per kilometer, but the distributions still exhibit significant variation across rider types. Figure 9 reveals that for rider type 4, the charged prices during the afternoon peak hours (from 16:00 to 18:00) are significantly higher than those for other time slots. This observation suggests an over-demand tendency during the afternoon peak hours.

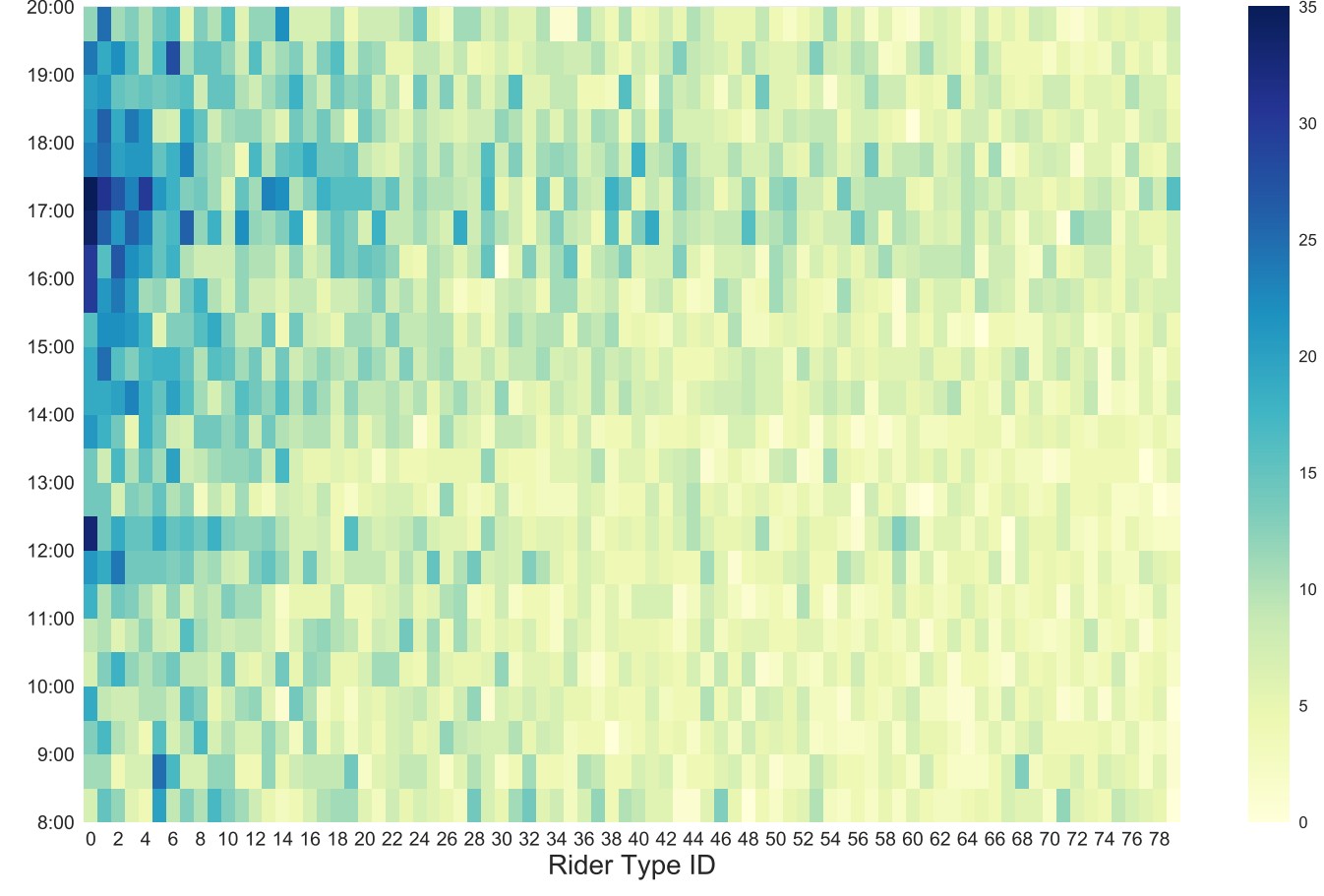

**Figure 8: The distribution of the total number of arrivals for all rider types at various times on May 5, 2017, which suggests a peak-hour trend between 16:00 and 18:00 when most rider types experience their highest arrival rates.**

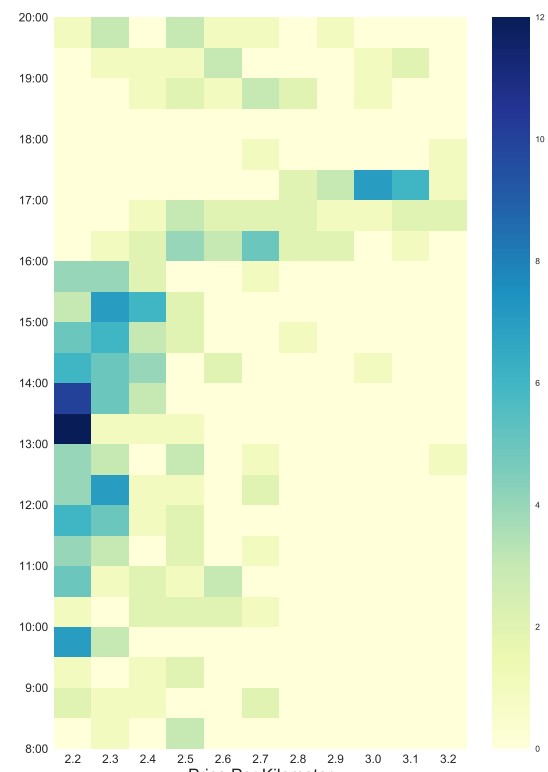

Figure 9: The number of trip records for rider type 4 from 8:00 to 20:00. The charged prices during the afternoon peak hours (from 16:00 to 18:00) are significantly higher than those for other time slots.

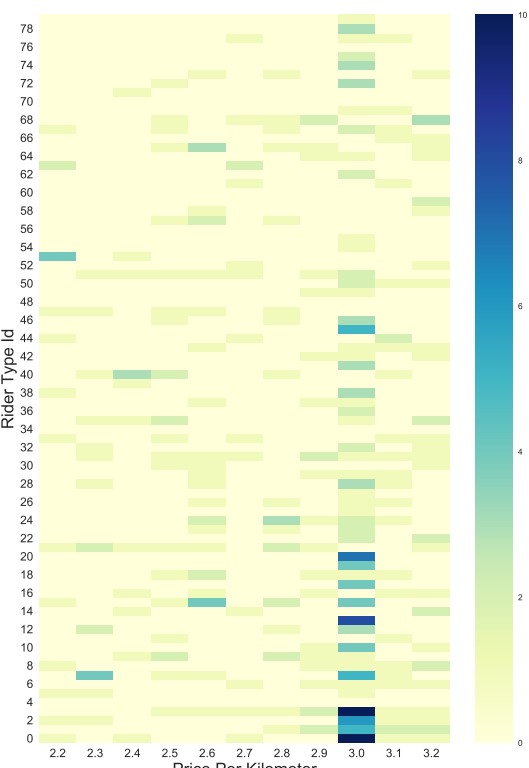

Figure 10: The number of trip records for all rider types between 16:00 and 18:00.

