# OpenReview forum: "Tight Competitive and Variance Analyses of Matching Policies in Gig Platforms"
_ACM.org/TheWebConf/2024/Conference — TheWebConf24 Oral_

### Official Review · Reviewer_f5dN · 2023-11-21

**Novelty:** 4
**Technical Quality:** 6

**Review:**

# Summary

This paper studies the prophet inequality matching with vertex arrival under non-identical distributions. The authors analyze two algorithms -- ATT($\gamma$) and SAMP($\gamma$) with pre-specified parameter $\gamma$ by characterizing the tight competitive ratios, and variance of the matching size. In particular, algorithm ATT(1/2) attains the optimal competitive ratio.

# Strengths

This paper studies a well-motivated model and analyzes two natural algorithms. Unlike most results in the online algorithm design, this paper not only characterizes the competitive ratios but also the variance. All the results are tight. Algorithm ATT(1/2) attains the optimal competitive ratio. The paper is well-written and easy to ready.

# Weaknesses

My main concern is the novelty of the paper, both theoretically and conceptually: (i) The optimal competitive ratio of 1/2 for prophet inequality matching is known and can be achieved by various algorithms [AHL-12][Ala-14][FGL-14]. Though this paper considers a variant model where the algorithm also decides a price (aka assignment), the competitive ratio analysis follows from prior work without significant modification. (ii) The idea of analyzing the variance of randomized online algorithms is new but also appears in recent prior work [41][DILMV-23]. The variance result feels a little incomplete since it only characterizes the variance of the matching size, instead of the variance of total collected revenue.

# Comments

- Since this model generalizes the single-item prophet inequality, the 1/2-competitive upper bound of single-item prophet inequality naturally holds in this model. It would be helpful to clarify this in Section 2.2.

- For the upper of ATT($\gamma$) in Section 2.4, I do not see the necessity to consider general $m$, rather than $m = 1$ (i.e., single agent, single type, single assignment, and single time period).




*[FGL-14] Feldman, M., Gravin, N., & Lucier, B. (2014, December). Combinatorial auctions via posted prices. In Proceedings of the twenty-sixth annual ACM-SIAM symposium on Discrete algorithms (pp. 123-135). Society for Industrial and Applied Mathematics.*

*[Ala-14] Alaei, S. (2014). Bayesian combinatorial auctions: Expanding single buyer mechanisms to many buyers. SIAM Journal on Computing, 43(2), 930-972.*

*[AHL-12] Alaei, S., Hajiaghayi, M., & Liaghat, V. (2012, June). Online prophet-inequality matching with applications to ad allocation. In Proceedings of the 13th ACM Conference on Electronic Commerce (pp. 18-35).*

*[DILMV-23] Dinitz, M., Im, S., Lavastida, T., Moseley, B., & Vassilvitskii, S. (2023). Controlling Tail Risk in Online Ski-Rental. arXiv preprint arXiv:2308.05067.*

**Questions:**

I asked some questions/comments above but please don't feel obligated to respond.

**Reviewer Confidence:**

3: The reviewer is confident but not certain that the evaluation is correct

**Scope:**

4: The work is relevant to the Web and to the track, and is of broad interest to the community

---

### Official Review · Reviewer_PP1y · 2023-11-23

**Novelty:** 5
**Technical Quality:** 5

**Review:**

The paper considers a generalization of online matching with known distributions. Both matching and pricing issues are tackled. More precisely, at the arrival of an online vertex, one may offer an offline vertex as well as a price for being matched to this vertex, and the online vertex may accept or reject the offer with probability depending on the price. If the online vertex accepts the offer, then it will be matched, and one can make a profit. The goal is to maximize the expected total profit. Two algorithms are proposed, and their competitive ratios and variances are analyzed.  Let $B$ be the number of offline vertices. The first algorithm has a competitive ratio of $\gamma$ and a variance of $\gamma*(1 - \gamma)*B$ for any $\gamma \in [0, 1/2]$. The second algorithm has a competitive ratio of $\gamma*(1- \gamma)$ and a variance of $\beta * (1 -\beta) *B$ where $\beta= \min(\gamma, 1/2)$. The analysis of both algorithms is tight.

Their algorithms are, in my opinion, an extension of those in a previous paper (Reference [41]) where only matching issues are considered. Indeed, the algorithms in these two papers look quite similar at first sight. But after reading the analysis, I find that this extension is not as trivial as it looks since the model in this paper is more complex than that in the previous paper. In particular, in the ATT algorithm, the debilitation factor should be carefully chosen in order to tackle with the situation that an online vertex may reject an offer.

The overall writing quality is good, and it is easy to follow this paper.

Other Comments:

In theoretical computer science, there is a long line of research on online (stochastic) matching, which is highly related to this paper. I think the authors should cite these results.

**Questions:**

none

**Reviewer Confidence:**

3: The reviewer is confident but not certain that the evaluation is correct

**Scope:**

4: The work is relevant to the Web and to the track, and is of broad interest to the community

---

### Official Review · Reviewer_kCuG · 2023-11-24

**Novelty:** 6
**Technical Quality:** 7

**Review:**

This paper provides competitive ratios and variances of simple LP-based matching policies upon "known adversarial distributions" (MP-KAD problems). The setting consists of constantly present offline agents $i$ and randomly arriving online agents $j$. At each time step, exactly one online agent $j$ arrives (w.p. $q_{j,t}$, and the decision maker (matching platform) has to match it with an available offline $i$ (depending on feasibility constraints $(i,j) \in E$ as well as whether $i$ still has capacity to take $j$) or reject $j$. Then, online agent $j$ decides whether to accept the assignment (w.p. $p_{f,t}$). The objective is to maximize total reward, where each accepted assignment $f$ at time $t$ yields a reward $w_{f,t}$. The authors first give a benchmark LP whose optimal objective value is an upper bound of a clairvoyant optimal value. Then, the authors provide two sampling algorithms using this LP and establish their competitive ratios.

For ATT$(\gamma)$, $\gamma \in [0,1/2]$, the competitive ratio is $\gamma$; there exists a problem instance where ATT$(\gamma)$'s CR is exactly $\gamma$. For $\gamma = 1/2$, ATT$(1/2)$ achieves a $1/2$ CR, which is optimal. In terms of variance analysis, for the number of successful assignment $H$, the authors show that its variance is bounded by $\gamma(1-\gamma)B$ where $B$ is the total number of offline agents (total budget due to the w.l.o.g. unit budget assumption). The main idea of the proof is to establish that $H_i$'s (indicator of whether offline agent $i$ ever gets a successful assignment) are negatively associated, which implies the variance of their sum is bounded by the sum of their variances. Furthermore, it can be shown that this bound is tight for ATT$(\gamma)$, $\gamma \in [0,1/2]$; there exists an instance where ATT$(\gamma)$ has CR exactly $\gamma$ and variance exactly $\gamma(1-\gamma)B$.

Similar results are established for the other algorithm SAMP$(\gamma)$. Both the CR and variance bounds are tight for all $\gamma \in [0,1]$.

**Questions:**

- Regarding the acceptance/rejection decision after the algorithm chooses an assignment (matching $i$ with $j$ and picking a price $a_k$), does it matter whether $i$ or $j$ makes the acceptance/rejection decision? Since it is a Bernoulli r.v. independent of the algorithm's choice, it should not matter? In other words, in the current problem setting, acceptance/rejection decision can be made by $i$, $j$, or entirely from the environment's randomness? It might be worth pointing this out. And do $X_{j,t}$, $Y_{f,t}$, $Z_{f,t}$ need to be independent, or just uncorrelated? Based on the proofs (e.g., Line 762), it seems the latter (or even weaker assumptions) will suffice.
- Is the fixed, known set of prices needed? In other words, does allowing a set of prices for each assignments makes the problem more general and/or harder? In other words, can the problem setting be reduced to a single type for each edge $(i,j)$ by having $k$ copies of $i$ (or $j$)? Can you show that the current setting is essentially different from the said simpler setting?
- If the current setting is indeed more general than the single-edge-type setting, can it be (easily) extended to cover the setting of broadcasting $j$ to multiple offline agents $i$ as well? The latter provides more modeling flexibility as certain ride-sharing platforms already do this: the first driver within the broadcast accepting the request (with price) gets it; if no driver accepts it, then it forfeits. This may bring more applicability to the current problem setting.
- The term "adversarial", although appeared in previous literature, may be misleading: neither the arrival distributions or actual arrivals are chosen by an adversary. I'd suggest naming this setting as "nonstationary", "nonuniform", or "heterogeneous" distributions, unless the authors have a strong reason for using it.
- [nit] Important notations (e.g., $B$) are left out in the table; some unimportant ones (e.g., $G$) may be removed/replaced instead. Might also worth mentioning the overall (w.l.o.g.) assumption $b_i=1$ in the table or in a numbered "Assumption" item (a few assumptions used throughout the analysis).
- [nit] "Both claims here are irrespective of the benchmark LP." seems redundant in Lemma statements and Figure captions. Having it in "Remarks on Theorems 1 and 2" and other text is already sufficient.

**Ethics Review Description:**

N.A.

**Reviewer Confidence:**

3: The reviewer is confident but not certain that the evaluation is correct

**Scope:**

4: The work is relevant to the Web and to the track, and is of broad interest to the community

---

### Official Review · Reviewer_5n5X · 2023-12-02

**Novelty:** 6
**Technical Quality:** 5

**Review:**

The paper provides an online-matching-based
model for matching and pricing, which is motivated primarily by real-world gig platforms like ride-hailing, crowdsourcing, and online recommendations.
It focuses on two parametrized LP-based sampling policies (ATT and SAMP) and provides competitive and variance analyses for these resulting in tight competitive ratios.

I think the results are nice and the techniques quite broadly applicable. The paper does a good job comparing with its direct predecessor, the work of Xu (Exploring the Tradeoff between Competitive Ratio and Variance in Online-Matching Markets).

**Questions:**

-

**Reviewer Confidence:**

3: The reviewer is confident but not certain that the evaluation is correct

**Scope:**

4: The work is relevant to the Web and to the track, and is of broad interest to the community

---

### Decision · Program_Chairs · 2024-01-22

**Decision:**

Accept (Oral)

**Comment:**

Strengths:
 + One of the earlier (but not the first) papers studying variance bounds in online matching, an interesting and lesser-studied topic
 + Provides non-trivial technical contributions with tight bounds
 + Well-written and easy to read

 Weakness:
 - Largely an extension / direct follow-up to Xu et al [41], which reduces the conceptual and technical novelty of the work.
 - Insufficient engagement with large literature on online stochastic matching

 Overall, this paper provides a solid technical step toward better understanding variance bounds in online matching, and how to tune online matching algorithms to obtain both good competitive ratio and variance bounds. While neither the question of studying variance nor the types of algorithms proposed are novel, the results are nice, and analysis is sound, non-trivial, and requires some technical nuance and tuning of algorithm inputs (debilitation factor) to obtain the desired bounds.